# A Comparative Analysis between Whole Chinese Yam and Peeled Chinese Yam: Their Hypolipidemic Effects via Modulation of Gut Microbiome in High-Fat Diet-Fed Mice

**DOI:** 10.3390/nu16070977

**Published:** 2024-03-27

**Authors:** Qiqian Feng, Jinquan Lin, Zhitao Niu, Tong Wu, Qun Shen, Dianzhi Hou, Sumei Zhou

**Affiliations:** 1Beijing Advanced Innovation Center for Food Nutrition and Human Health, School of Food and Health, Beijing Technology and Business University, Beijing 100048, China; fengqiqian2022@163.com (Q.F.); 13160537598@163.com (J.L.); a824630407@163.com (Z.N.); zhousumei@btbu.edu.cn (S.Z.); 2Beijing Engineering and Technology Research Center of Food Additives, School of Food and Health, Beijing Technology and Business University, Beijing 100048, China; 3Key Laboratory of Green Manufacturing and Biosynthesis of Food Bioactive Substances, China General Chamber of Commerce, Beijing 100048, China; 4College of Food Science and Nutritional Engineering, China Agricultural University, Beijing 100083, China; estela_tong@163.com (T.W.); shenqun@cau.edu.cn (Q.S.)

**Keywords:** whole Chinese yam, peeling treatment, hyperlipidemia, lipid disorders, gut microbiota

## Abstract

Chinese yam is a “medicine food homology” food with medical properties, but little is known about its health benefits on hyperlipidemia. Furthermore, the effect of peeling processing on the efficacy of Chinese yam is still unclear. In this study, the improvement effects of whole Chinese yam (WY) and peeled Chinese yam (PY) on high-fat-diet (HFD)-induced hyperlipidemic mice were explored by evaluating the changes in physiological, biochemical, and histological parameters, and their modulatory effects on gut microbiota were further illustrated. The results show that both WY and PY could significantly attenuate the HFD-induced obesity phenotype, accompanied by the mitigative effect on epididymis adipose damage and hepatic tissue injury. Except for the ameliorative effect on TG, PY retained the beneficial effects of WY on hyperlipemia. Furthermore, 16S rRNA sequencing revealed that WY and PY reshaped the gut microbiota composition, especially the bloom of several beneficial bacterial strains (*Akkermansia*, *Bifidobacterium*, and *Faecalibaculum*) and the reduction in some HFD-dependent taxa (*Mucispirillum*, *Coriobacteriaceae_UCG-002*, and *Candidatus_Saccharimonas*). PICRUSt analysis showed that WY and PY could significantly regulate lipid transport and metabolism-related pathways. These findings suggest that Chinese yam can alleviate hyperlipidemia via the modulation of the gut microbiome, and peeling treatment had less of an effect on the lipid-lowering efficacy of yam.

## 1. Introduction

The prevalence rate of hyperlipidemia has shown a sharp rise worldwide in recent years [1,2], which is usually caused by significant changes in consumers’ dietary habits, especially the long-term intake of fatty foods. Hyperlipidemia is a chronic disease typically characterized by abnormal lipid metabolism, which might trigger the development of most cardiovascular and cerebrovascular diseases such as diabetes mellitus, hypertension, and atherosclerosis [3,4]. Persuasive evidence suggests that gut microbiota, as a symbiotic microbial community within the human body, are intrinsically linked to nutrient absorption, digestion, and lipid metabolism [5]. Hyperlipidemic sufferers are thought to have disturbed gut metabolism as compared to the normal population [6]. Animal studies also show that high-fat-diet (HFD)-induced hyperlipidemic mice had a negatively affected relative abundance of gut microbiota composition, thus disrupting the intestinal barrier and homeostasis [7,8]. The conclusion that functional foods or their bioactive ingredients can effectively regulate the composition of gut microbiota, contributing to the improvement of lipid metabolism, has been widely confirmed [9,10,11]. Therefore, targeting the modulation of gut microbiota via dietary intervention might be an effective approach to combat HFD-induced hyperlipidemia.

Yam, also known as “Shanyao” in China, is classified botanically as belonging to the genus *Dioscorea* in the family *Dioscoreaceae*, which is capable of forming plump tubers [12]. As an important source of various nutrients and phytochemicals, yam is widely cultivated in Africa, Asia, Oceania, and South America [13]. According to the ancient Chinese pharmacopoeia “Compendium of Materia Medica”, yam was recorded to be conducive to nourishing the spleen and stomach, strengthening immunity, and delaying ageing. Furthermore, emerging evidence has demonstrated that the bioactive ingredients of yam possess many other potential health benefits such as antioxidant [14], anti-inflammatory [15], anti-tumor [16], anti-diabetic [17], and anti-obesity and hypolipidemic [18] activities. However, the research lipid-lowering activity is frequently limited to studies on single components or extracts of yam [19,20], and it is unclear whether yam maintains similar effects when consumed as a whole food. Moreover, some functional components (e.g., polysaccharides or resistant starch) in yam are indigestible or unabsorbed in the upper gastrointestinal tract, and therefore can reach the gut intact and interact with gut microbiota. This study has validated the hypolipidemic efficacy of yam when consumed as a whole food, and explored the modulatory effects of yam on the gut microbiota.

Whole Chinese yam and peeled Chinese yam are the two forms of daily consumption, but the latter seems to be more popular due to the removal of the numbing effects and astringency of Chinese yam peel. Noticeably, compared with the yam flesh, the total phenolic and total flavonoid contents were higher in the yam peel, which might contribute to their biological activities [21]. In this case, the contents of bioactive compounds in yam will be inevitably changed after peeling processing, but whether this will affect the lipid-lowering activity of yam is unknown. Therefore, for whole Chinese yam and peeled Chinese yam, their different hypolipidemic effects and the regulation of gut microbiota remain to be further clarified.

In the light of the above considerations, we determined the physiological, biochemical, and histological indices of HFD-induced mice after yam intake and analyzed the alterations in gut microbial composition in this study. Moreover, a comparative analysis between whole Chinese yam and peeled Chinese yam was conducted to elucidate their different hypolipidemic effects. The results help us to better understand the lipid-lowering activity of Chinese yam and the effect of peeling treatment on its efficacy.

## 2. Materials and Methods

### 2.1. Preparation of Whole Chinese Yam and Peeled Chinese Yam

The Chinese yam tubers (*Dioscorea* spp.) used in this study were purchased from Jiaozuo City, Henan Province, China. The washed whole Chinese yam tubers were cooked for 40 min in a steam chamber, and then peeled Chinese yam tubers were obtained by manually peeling the steamed whole Chinese yam tubers. The cooked WY and PY were freeze-dried, followed by pulverization and sieving (80 mesh). The yam powder was stored at −20 °C under refrigeration for subsequent use. The nutritional compositions of WY and PY are shown in Appendix A.

### 2.2. Animal Experimental Design and Diets

Six-week-old C57BL/6J male mice weighing 20 ± 2 g were purchased from Beijing Vital River Laboratory Animal Technology Co., Ltd. (Beijing, China). Four mice per cage were maintained in a standard pathogen-free (SPF) facility with controlled conditions (23 ± 2 °C, 55 ± 10% humidity, 12 h light–dark cycle) and free availability of food and water. The detailed animal experimental procedure is illustrated in Figure 1A. All animal experimental protocols in this study complied with the ethical guidelines of the National Research Council guidelines and were endorsed by the Institutional Animal Care and Use Committee of the China Agricultural University (AW90303202-5-1).

After one week of acclimation, mice were randomly divided into 4 groups (8 mice in each group) for subsequent experiments: (1) normal control diet (NCD) group (D12450, 10% energy from fat, 3.85 total kcal g^−1^); (2) high-fat diet (HFD) group (D12492, 60% energy from fat, 5.24 total kcal g^−1^); (3) whole Chinese yam group (HFD-WY, HFD supplemented with 25% WY, 60% energy from fat, 5.24 total kcal g^−1^; (4) peeled Chinese yam group (HFD-PY, HFD supplemented with 25% PY, 60% energy from fat, 5.24 total kcal g^−1^). In order to successfully establish a hyperlipidemic mice model, all mice were fed for 12 weeks on a defined diet. The diets were provided by Changzhou Shuyishuer Bio-Tec Co., Ltd. The dosage levels of WY and PY in diets were selected based on a previous published study [22]. The specific compositions of the experimental diets are presented in Appendix A.

### 2.3. Sample Collection

Food intake and body weight were measured weekly over the course of the 12-week intervention. At the end of the experiment, mice were fasted for 12 h. Blood samples from the orbital vascular plexus were collected and centrifuged (1300× *g*, 4 °C, 10 min), which was stored at −80 °C. Following cervical dislocation to execute the mice, the white adipose tissue (WAT) was weighed. In addition, parts of the liver samples and epididymal adipose tissues were fixed in 4% paraformaldehyde.

### 2.4. Serum Biochemical Analysis

Serum levels of triglycerides (TG), total cholesterol (TC), low-density lipoprotein cholesterol (LDL-C) and high-density lipoprotein cholesterol (HDL-C), and activities of alanine aminotransferase (ALT) and aspartate aminotransferase (AST), were measured by a 3100 automatic biochemical analyzer (Hitachi Ltd., Tokyo, Japan).

### 2.5. Histopathological Analysis

The liver and adipose tissues fixed in 4% paraformaldehyde were dehydrated, embedded in paraffin, and cut into 5 μm sections. Staining was finally performed with hematoxylin and eosin (H&E). Samples were viewed under a Nikon Eclipse E100 microscope (Nikon, Tokyo, Japan) and photographs were collected. Adipose tissue sections from each mouse were analyzed using ImageJ software (version 2.3.21, National Institutes of Health, NIH) to measure adipocyte area.

### 2.6. Gut Microbiota Analysis

Fecal samples from mice were collected in sterile test tubes at week 12 of the experiment, which were immediately frozen in liquid nitrogen and stored at −80 °C. DNA was extracted from the samples according to the E.Z.N.A. Soil DNA kit. Nanodrop spectrophotometry and 1% agarose gel electrophoresis were applied to determine the DNA concentration and purity, as well as integrity, respectively.

Based on the V3-V4 hypervariable region of 16S rRNA, the extracted genomic DNA was amplified by polymerase chain reaction (PCR) using primer pairs 338F (5′-ACTCCTACGGGGAGGCAGCAG-3′) and 806R (5′-GGACTACHVGGGTWTCTAAT-3′). The detailed reaction conditions were followed as previously reported [23]. Subsequently, the obtained 16S rRNA gene amplicons were extracted, purified, and quantified. The MiseqPE300 platform was used for sequencing the purified amplicons (Shanghai Meiji Biomedical Technology Co., Ltd., Shanghai, China) and libraries were constructed according to the Illumina MiSeq platform (Illumina, San Diego, CA, USA).

High-quality sequences were screened by quality filtering for bioinformatics analysis. Operational taxonomic units (OTUs) were clustered in Quantitative Insights Into Microbial Ecology (QIIME) software (version 1.9.1) (similarity set to 97%), and samples were analyzed for α-diversity and β-diversity using Ribosomal Database Project (RDP) classification software. The relative abundance of bacterial taxa in fecal samples at each level was analyzed by QIIME software (http://qiime.org/install/index.html, accessed on 3 April 2023) [24]. Linear discriminant analysis (LDA) was performed by Linear discriminant analysis Effect Size (LEfSe) software (https://huttenhower.sph.harvard.edu/lefse, accessed on 3 April 2023) to filter for taxonomic units that yielded significant differences between groups at the genus level (*p* < 0.05 with LDA values > 3.0) [23]. 16S functional prediction analysis was performed by PICRUSt (version 1.1.0) [25].

### 2.7. Statistical Analysis

All data are expressed as mean ± SEM (standard errors of the mean). Statistical analyses were performed using SPSS statistical software (version 22.0, IBM Corporation, Chicago, IL, USA), and Tukey’s test for multiple comparisons in one-way analysis of variance (ANOVA) was used to evaluate the significant differences between groups. The results were statistically significant with a default of *p* < 0.05.

## 3. Results

### 3.1. The Effects of WY and PY on Body Weight, Food Intake, and White Adipose Tissues in Mice

To investigate the effects of WY and PY on the obesity phenotypes, mice were separately raised on HFD with or without WY and PY supplementation for 12 weeks (Figure 1A). After 12 weeks of feeding, the body weights of the HFD group were significantly higher than those of the NCD group (Figure 1B, *p* < 0.05). In addition, the body weight gain, WAT weight, and energy intake in the HFD group showed a significant increase compared with the NCD group (Figure 1C,D,F, *p* < 0.05), while WY and PY supplementations were effective in reducing total body weight, body weight gain, and WAT weight in HFD-fed mice without affecting food intake (Figure 1B–E). No significant differences were observed in these indices between the HFD-WY and HFD-PY groups. In terms of various phenotypic indicators, WY and PY exhibited similar beneficial effects in HFD-fed mice.

### 3.2. The Effects of WY and PY on Serum Lipid Profiles

HFD can trigger hyperlipidemia by inducing abnormalities in lipid metabolism, as evidenced by the significantly increased TC, TG, and LDL-C of mice in the HFD group (Figure 2) [26]. Due to an adaptive response to a greater need to transport the increased lipids, the HDL-C level in the HFD group showed a significant increase. The intakes of both WY and PY significantly reduced the elevated TC, LDL-C, and HDL-C levels (Figure 2A,D, *p* < 0.05) in mice fed with HFD, but the TG levels were only significantly decreased in the HFD-WY group (Figure 2B, *p* < 0.05). Moreover, significant differences were observed in the serum lipid profile levels between the HFD-WY and HFD-PY groups.

### 3.3. The Effects of WY and PY on the WAT Hypertrophy and Hepatic Steatosis

As shown in Figure 3A, compared with the NCD group, the adipocytes in the HFD group were apparently enlarged in size and loosely arranged in a disordered manner, whereas the adipocytes tended to be uniformly arranged after the WY and PY consumption. Furthermore, the adipocyte area was significantly lower in the HFD-WY and HFD-PY groups than in the HFD group (Figure 3C, *p* < 0.05). The liver histological examination of mice revealed that, in the NCD group, hepatocytes around the central vein were arranged radially with a regular and tight distribution (Figure 3B). On the contrary, hepatocytes in the HFD group underwent multiple cycles of cytoplasmic vacuolization and marked hepatic steatosis, indicating that liver tissue was damaged. Following the WY and PY interventions, liver tissue damage was notably ameliorated, with increased similarity to the NCD group, which was supported by the significantly lowered liver steatosis score (Figure 3D, *p* < 0.05). In addition, the levels of liver injury biomarkers (ALT and AST) were determined. The results show that ALT and AST levels in the HFD-WY and HFD-PY groups were significantly lower than those in the HFD group, and WY was significantly more capable than PY of reducing the AST level (Figure 3E, *p* < 0.05). The above evidence proves that WY and PY could significantly attenuate HFD-induced adipocyte abnormalities and liver injury.

### 3.4. The Effects of WY and PY on the Gut Microbiota Composition

The Sobs index (dilution curve) curve and the Shannon–Wiener curve flatten out along the right end, indicating the reasonableness and completeness of the sequencing data (Appendix A). α-diversity analysis was applied to characterize the abundance and diversity of microbial communities. As shown in Figure 4A, there were no significant differences in the ACE and Chao indexes between the HFD group and HFD-WY and HFD-PY groups. According to the Shannon and Simpson indexes, the HFD group exhibited the highest community diversity, and the HFD-WY and HFD-PY groups were not significantly different from the NCD group (Figure 4A). Principal coordinate analysis (PCoA) reflects the β-diversity of the gut microbiota. There was a significant separation between the HFD and NCD groups, suggesting that HFD caused remarkable changes in the gut microbiota community of mice (Figure 4B). Fractional clustering was observed in the HFD-WY and HFD-PY groups, implying the existence of partially similar gut microbiota compositions in the two groups. However, the HFD-WY and HFD-PY groups were significantly distant from the HFD group. Figure 4C shows that the HFD-WY and HFD-PY groups were the first to show intergroup similarity and clustered with the NCD group, but the HFD group was significantly separated from the other groups. Collectively, this evidence suggests that WY and PY supplementation might prevent the gut microbiota disorder caused by HFD.

In order to evaluate specific microbial populations in mice under WY and PY consumption, the relative abundances of different groups were analyzed at the phylum and genus levels. At the phylum level, all groups were dominated by *Firmicutes*, *Actinobacteria*, and *Bacteroidota* (Figure 5A). *Firmicutes* and *Bacteroidota* affected the organism’s intake and catabolism of dietary nutrients, thus the ratio of *Firmicutes* to *Bacteroidota* (F/B) was regarded as a crude and simple indicator of flora monitoring. The F/B ratio was significantly higher in the HFD group compared to the NCD group. However, the interventions of WY and PY significantly prevented the elevated F/B ratio caused by HFD, but there was no significant difference in their effects (Figure 5B, *p* < 0.05).

Heat maps were constructed to further analyze the details of the gut microbiota in different groups (Figure 5C). In total, 30 OTUs with outstanding relative abundance were screened to identify key populations. HFD significantly increased 18 OTUs and decreased 11 OTUs compared to the NCD group. WY or PY supplementation remarkably reversed the changes of 21 and 23 OTUs, respectively. Moreover, supplementation with WY and PY significantly enhanced the relative abundances of *Akkermansia*, *Bifidobacterium*, and *Faecalibaculum*, and reduced the relative abundance of *Mucispirillum*, *Coriobacteriaceae_UCG-002*, and *Candidatus_Saccharimonas* (Figure 5D,E). Noticeably, PY was more effective than WY in increasing beneficial bacteria such as *Akkermansia* and *Bifidobacterium* levels, and decreasing *Candidatus_Saccharimonas* levels.

### 3.5. Key Phylotypes of Gut Microbiota Responding to WY and PY Supplementation

LDA performed by LEfSe clarified the dominant bacterial community in each group. We have further demonstrated significant differences in specific bacterial taxa in the HFD group after 12 weeks of WY and PY interventions from different taxonomic systems. As shown in Figure 6, at the phylum level, the HFD group was enriched by dramatic increases in the *Coriobacteriaceae_UCG_002* and *Mucispirillum*, which may be main indicators of abnormal lipid metabolism. In addition, the characteristic bacterial genera in the HFD-WY and HFD-PY groups included *Faecalibaculum*, *Bifidobacterium*, and *Akkermansia*, which is consistent with the above results.

### 3.6. Metabolic Pathways Altered by WY and PY Supplementation in the Gut Microbiome

Changes in the gut microbiota composition frequently lead to variations in metabolic pathways and functions. Therefore, 16S functional prediction analyses were performed by PICRUSt2 analysis and Clusters of Orthologous Genes (COG) database comparisons (Figure 7). The results show that WY and PY supplementation significantly affected 10 and 12 metabolic pathways, respectively. They jointly up-regulated amino acid and carbohydrate transport and metabolism. In addition, PY supplementation has a remarkable regulatory effect on lipid transport and metabolism, while energy production and conversion were significantly enriched in the HFD-WY group.

## 4. Discussion

Hyperlipidemia can trigger a series of complications such as coronary heart disease and atherosclerosis, which is a risk factor threatening human health. To date, the single component of yam (e.g., polysaccharides and resistant starch) has been reported to alleviate hyperlipidemia [18,27]. Noticeably, it has demonstrated that whole food, as a more accepted dietary intervention for individuals with hyperlipidemia, may exert its health benefits through the synergistic effects of nutrients combined with non-nutrient compounds. However, the effects of whole Chinese yam on hyperlipidemia have been rarely published, especially from the perspective of gut microbiota. In addition, considering the demands of different individuals as regards the taste of yam, a comparative analysis between whole Chinese yam and peeled Chinese yam was conducted to evaluate their differences as regards the regulatory role of lipid metabolism. 

In the present study, the results indicate that WY and PY significantly reduced obesity phenotype and serum TC and LDL-C levels, accompanied by the improvement effects of WAT hypertrophy and liver injury. Notably, in terms of lowering serum TG, LDL-C, and AST levels, the findings of the HFD-WY group are more significant than those of the HFD-PY group. In general, WY was capable of exerting lipid-lowering effects. Moreover, although the peeling treatment slightly affected the therapeutic effects of yam, PY was still able to show significant health benefits. The ability of Chinese yam to effectively alleviate abnormalities in lipid metabolism is inextricably linked to its own substantial nutrient and functional components. A significant difference was observed in dietary fiber content between whole yam and peeled yam (Appendix A). Dietary fiber is a natural component recognized for improving lipid levels in serum and liver [28]. In a previous study, the presence of dietary fiber in yam may have inhibited the formation of cholesterol micelles and promote the excretion of bile acids and cholesterol in rats, which in turn led to a significant reduction in TG levels [29]. Our findings similarly demonstrate that WY was more significant than PY in reducing TG and LDL-C levels due to the presence of dietary fiber. Resistant starch was regarded as another typical dietary fiber improving lipid composition and alleviating metabolic dysfunction induced by hyperlipidemia [18]. A previous study has also suggested that the intake of yam polysaccharides may inhibit excess adipose tissue accumulation [20]. As the main active compound present in yam, dioscin (which can be metabolized to diosgenin *in vivo*) possesses lipid-lowering and cholesterol-lowering properties [30]. In addition, phenolics are generally recognized as promoting health benefits. The phenolic compounds extracted from yam, such as 3,3′,5-trihydroxy-2′-methoxybibenzyl, have been proven to exhibit inhibitory activity against pancreatic lipase in vitro [31]. The total phenolics and total flavonoids in yam peel were higher than those in yam flesh [21], which may also explain the difference in efficacy between WY and PY. Overall, the lipid-lowering effects of WY and PY might be attributed to the synergistic effect of these bioactive compounds [32]. Furthermore, these functional components show poor bioavailability, and thus might regulate the composition of gut microbiota [33].

Symbiotic microorganisms in the human gut co-operatively contribute to the regulation of energy balance and immune system, and thus the gut microbiota have been accepted as another key indicator of lipid metabolism [34]. Studies have previously reported that supplementation with yam extract has a modulatory effect on intestinal disorders, thereby maintaining host health [35,36]. In this study, the results of α and β diversity show that the WY and PY supplementation significantly altered the diversity and abundance of gut microbiota in the HFD-fed mice and rendered them close to the normal group. The F/B ratio helps to rapidly and conveniently evaluate the gut microbiota for disorders. The gut microbiota of obese people were found to exhibit higher F/B ratios than healthy individuals [37,38]. WY and PY supplementation caused a significant reduction in F/B ratio in HFD-fed mice. Furthermore, the relationship between specific bacteria and biological benefits can be further elucidated by analyzing the genus level. *Akkermansia* and *Bifidobacterium* are considered “star genera” of gut microbiota, exhibiting beneficial effects on human health [39,40]. *Akkermansia* has been reported to strengthen the intestinal mucosal barrier, regulate the immune system, and reduce a series of HFD-induced negative effects [41,42]. *Bifidobacterium* could increase the content of acetate in the gut, which in turn plays a role in regulating fat accumulation [43]. *Akkermansia* and *Bifidobacterium* are highly correlated with the prevention of hyperlipidemia, which has been proven by numerous studies [44,45]. A previous study revealed that supplementation with whole mung beans could markedly improve lipid metabolism by increasing the relative abundance of *Akkermansia* and *Bifidobacterium* [23]. In addition, certain whole grains, such as Qingke, have been proven to combat obesity by modulating the interactions between gut microbiota and host lipid metabolism [46]. In our study, WY and PY supplementation led to a significant increase in the relative abundances of *Akkermansia* and *Bifidobacterium*, and the role of PY supplementation was even more apparent. Besides this, as another type of bacteria that promoted the production of SCFAs, *Faecalibaculum* levels in HFD-WY and HFD-PY groups were also significantly elevated. On the contrary, some bacteria whthatich have been proved to be positively associated with lipid levels or pro-inflammatory factors, such as *Mucispirillum*, *Coriobacteriaceae_UCG-002*, and *Candidatus_Saccharimonas* [47,48], were inhibited in the HFD-WY and HFD-PY groups. Unusually, PY was even more effective than WY in increasing the level of beneficial bacteria and decreasing the level of harmful bacteria. This may imply that yam flesh is mainly responsible for the regulation of gut microbiota. For example, resistant starch in yam (RSII), mainly present in yam flesh [49,50], is difficult to digest in the small intestine, and can improve lipid metabolism through modulating the gut microbiota [51]. Alterations in gut microbiota determine distinctions in bacterial community function. Significant modulations of amino acid and carbohydrate transport and metabolism were found in the HFD-WY and HFD-PY groups, while WY and PY were also each upgraded by energy production and conversion and lipid transport and metabolism, respectively. These results suggest that WY and PY can promote the colonic consumption of energetic substances to achieve energy balance via modulating the composition of gut microbiota. 

Based on the above analysis, it was found that the effects of WY and PY on lipid metabolism and gut microbiota were similar. The moderate peeling processing of yam has little effect on its own lipid-lowering efficacy, which is beneficial for the development of functional products. However, further explorations of the synergistic effects of nutrients and active ingredients in yam and their association with lipid-lowering are needed.

## 5. Conclusions

This study reveals the potential lipid-lowering activity of whole Chinese yam, as manifested by the reduction in body weight and white adipose tissue mass, the decrease in serum lipid levels, and the attenuation of epididymis adipose and liver tissue damage in HFD-fed mice. PY showed the same beneficial effects as WY in lowering blood lipids, with the exception of serum total cholesterol. The mechanism by which yam alleviated hyperlipidemia was associated with the reconstitution of the gut microbiota. Both WY and PY significantly improved the diversity and abundance of the gut microbiota and down-regulated F/B. Especially, at the genus level, PY was more significant than WY in increasing the relative abundance of beneficial bacterial strains (*Akkermansia* and *Bifidobacterium*) and normalizing the levels of HFD-dependent bacteria (*Candidatus_Saccharimonas*). In summary, our findings strongly suggest that yam can improve lipid metabolism by modulating gut microbiota, and that the peeling treatment preserves satisfactory lipid-lowering efficacy while enhancing the taste of yam.

## Figures and Tables

**Figure 1 nutrients-16-00977-f001:**
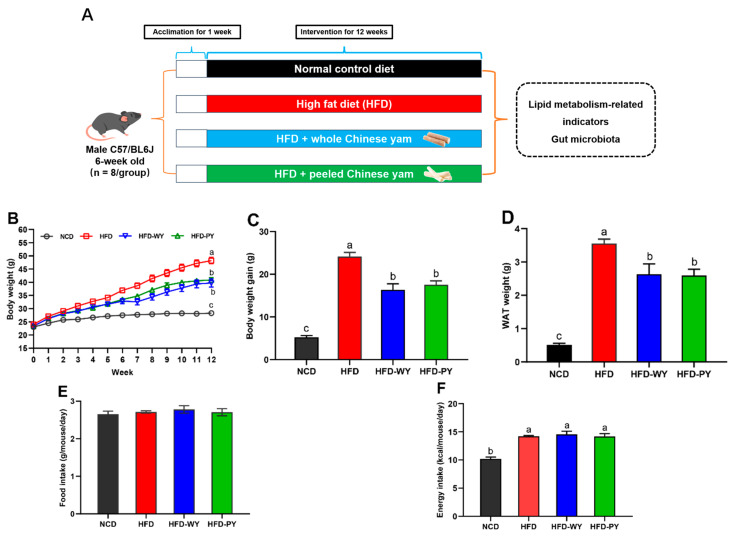
Effects of WY and PY supplementation on body weight, white adipose tissue (WAT) weight, and food and energy intake. (**A**) Specific experimental protocols. (**B**) Changes of body weight. (**C**) Body weight gain. (**D**) WAT weight. (**E**) Food intake. (**F**) Energy intake. Means with different letters on the bar charts represent significant differences (*p* < 0.05).

**Figure 2 nutrients-16-00977-f002:**
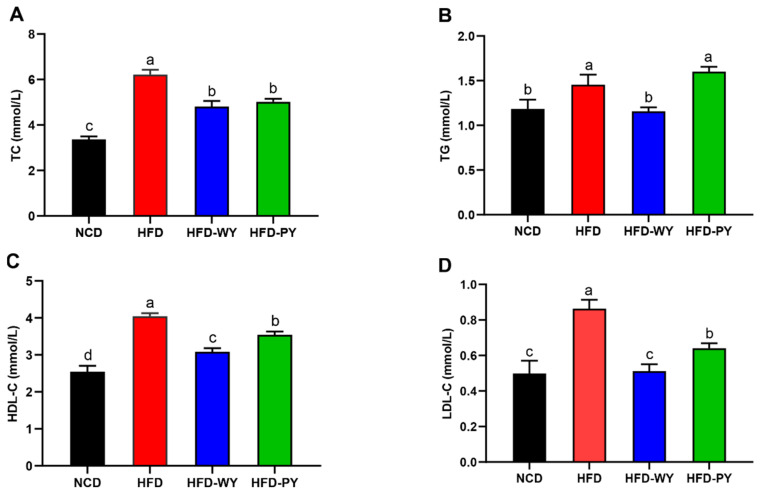
Effects of WY and PY supplementation on serum biochemical profiles. (**A**) TC. (**B**) TG. (**C**) HDL-C. (**D**) LDL-C. Means with different letters on the bar charts represent significant differences (*p* < 0.05).

**Figure 3 nutrients-16-00977-f003:**
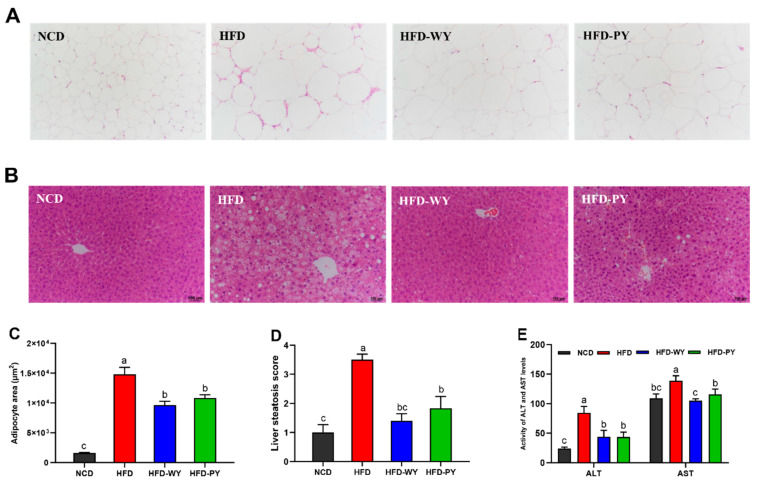
Effects of WY and PY supplementation on the WAT hypertrophy and hepatic steatosis. (**A**) Representative micrographs of H&E-stained epididymis adipose tissues (magnification 100×). (**B**) Representative micrographs of H&E-stained liver tissues (magnification 100×). (**C**) Adipocyte area. (**D**) Liver steatosis score (1+ = no change; 2+ = minor changes; 3+ = moderate changes; 4+ = severe changes). (**E**) The activities of serum ALT and AST. Means with different letters on the bar charts represent significant differences (*p* < 0.05).

**Figure 4 nutrients-16-00977-f004:**
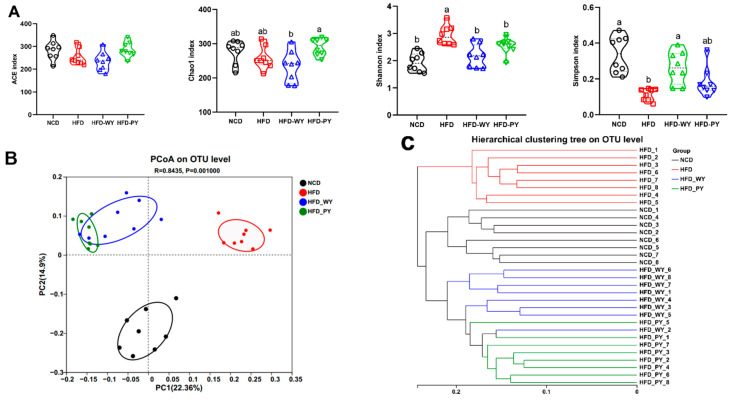
Effects of WY and PY supplementation on the α and β diversities of gut microbiota. (**A**) α diversity. (**B**) PCoA score plot. (**C**) Hierarchical clustering based on the unweighted-unifrac distance matrix. Means with different letters on the charts represent significant differences (*p* < 0.05).

**Figure 5 nutrients-16-00977-f005:**
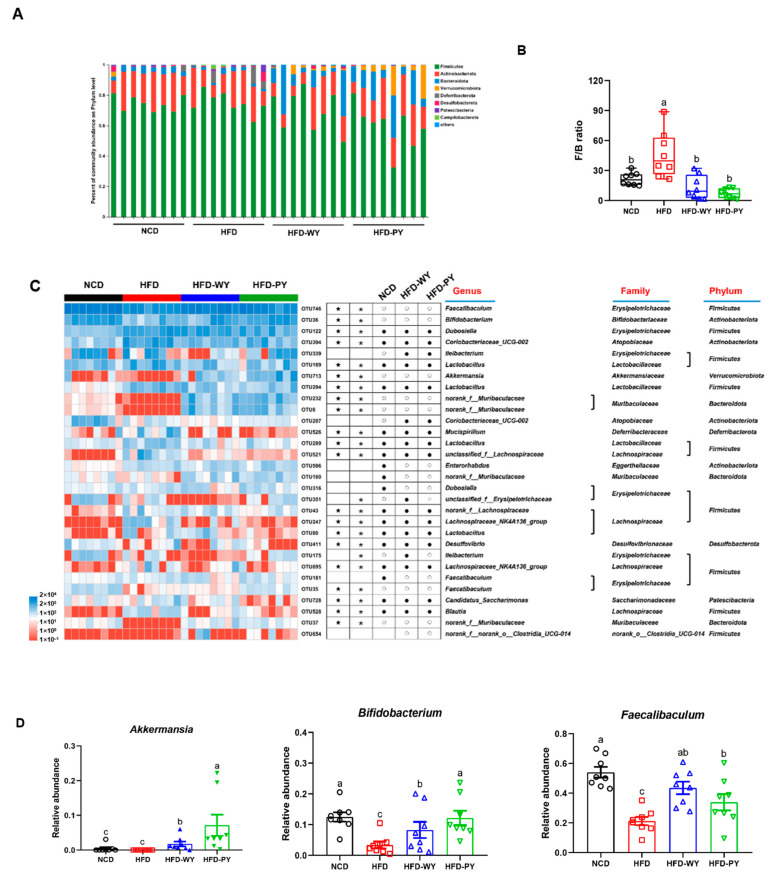
Modulating effects of WY and PY on gut microbial community in HFD-fed mice. (**A**) Relative abundance of the fecal microbiota at the phylum levels. (**B**) The F/B ratio. (**C**) Heatmap of the 30 OTUs altered by HFD responding to WY and PY supplementation. (**D**,**E**) The relative abundance of key genera. White circles (○) and black dots (●) represent OTUs whose relative abundance was higher and lower in the NCD group, HFD-WY group or HFD-PY group than in HFD group, respectively. Black stars (★) and asterisks (∗) represent OTUs whose relative abundance in the NCD group was altered by HFD and then reversed by WY and PY, respectively. Means with different letters on the bar charts represent significant differences (*p* < 0.05).

**Figure 6 nutrients-16-00977-f006:**
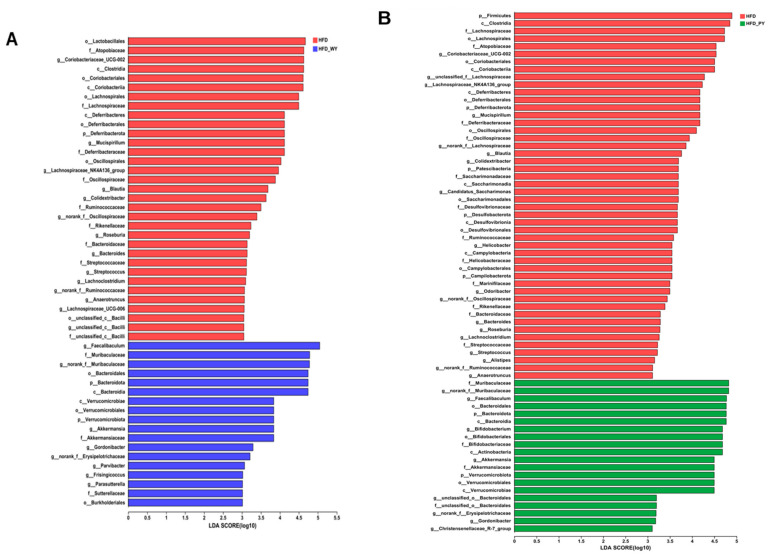
LDA scores obtained from LEfSe analyses, LDA score > 3.0. (**A**) HFD vs. HFD-WY. (**B**) HFD vs. HFD-PY.

**Figure 7 nutrients-16-00977-f007:**
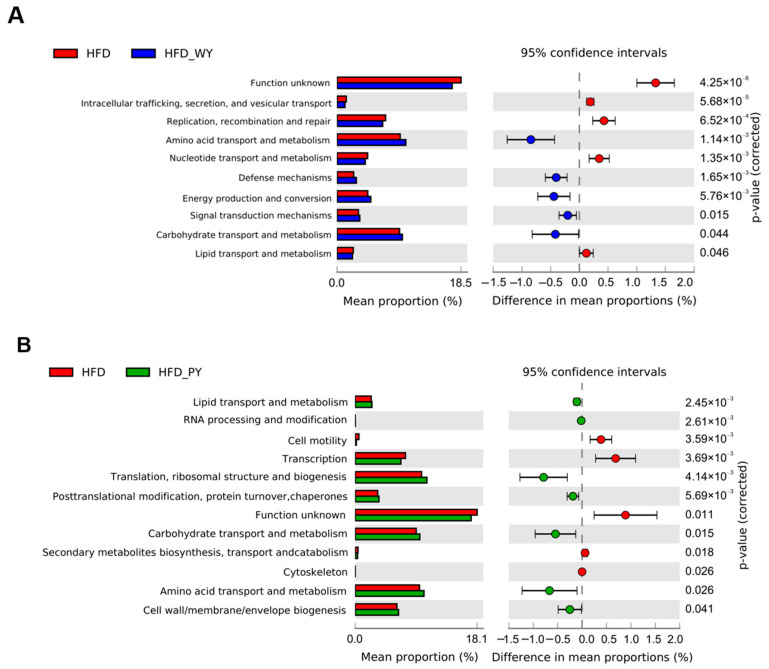
Predictive analysis of gut microbiota function in various groups of mice. (**A**) HFD vs. HFD-WY. (**B**) HFD vs. HFD-PY.

## Data Availability

The data presented in this study are available in this article.

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
