# Peer review of "A Comparative Analysis between Whole Chinese Yam and Peeled Chinese Yam: Their Hypolipidemic Effects via Modulation of Gut Microbiome in High-Fat Diet-Fed Mice"

_nutrients, 2024, doi:10.3390/nu16070977_

Round 1

Reviewer 1 Report

Comments and Suggestions for Authors

Thank you for submitting the manuscript "A comparative analysis between whole Chinese yam and peeled Chinese yam: their hypolipidemic effects via modulation of gut microbiome in high-fat diet-fed mice" to Nutrients. The research protocol included high-fat feeding of animals with forced and unpelled yam for 12 weeks. Although the number of animals in each group is lower than what we are used to in this type of study, especially for the control groups (where at least 12 animals have been used), the duration of the work was interesting (12 weeks). However, it would also be interesting for the authors to explain the reason for this duration for this product. Additionally, I have other considerations:

- Consider revising the entire text, adding definitions of abbreviations where appropriate.

- Line#40: this reference #4 was clearly included incorrectly as it did not study the effect of microbiota symbiosis on all these parameters. Please check.

- Line#57: if the lipidemic activity of yam has already been studied, it is necessary to justify the difference between this research and other studies already existing in the literature. Consider including at least one sentence on the topic.

- It needs to be clear in the introduction what the justification is for testing written yam and unpelled yam. Is there consumption in these two ways? How is it normally consumed? What is the percentage of consumers for each case?

- Much of the effects obtained may be related to the fiber composition of the yam, especially the difference between the driven and unpelled groups. I think it is very critical that this composition is added and compared in this work to explain why this behavior.

Reviewer 2 Report

Comments and Suggestions for Authors

Chinese yam is a plant well known in Chinese medicine, having a number of medicinal properties. Articles containing it in various forms are widely commercially available.

In the reviewed article, the authors decided to examine the difference in the activity of whole and peeled Chinese yam. However, it is not clear from the description whether they meant the whole plant or only the tuber of this plant.

The introduction to the article was written based on the latest literature, which is extremely rich and was selected appropriately.

The idea of comparing the whole and peeled form of Chinese yam is not entirely clear to me. Why do the authors decide to conduct such comparative studies, even though, as they write in the introduction, the peel contains a number of biologically active compounds (including flavonoids). Why didn't they include peelings in their research as a material potentially rich in active substances? The studies on the content of nutrients included in Table S1 do not prove the possibility of differences in the effects of whole and peeled forms of Chinese yam. The high carbohydrate content does not encourage its use in the diet of people with high lipid levels. The nutritional values of Chinese yam are widely known and available in the literature, and it would be good if the authors included literature references to these values.

The tests performed are typical and were planned and performed routinely. Their results were presented correctly and the resulting conclusions were described correctly. One may get the impression that the authors decided to conduct a series of studies according to the principle "let's see what will come out of it", without having any specific plan. Of course, this may just be my impression resulting from the lack of justification for the analyzes carried out. Results seem to indicate that the peeling process does not significantly affect the tests performed. The statement in the conclusion of the article that Chinese yam may be a source of food produced based on it may be surprising. The fact that Chinese yam can and even is a source of food has been known for a long time and no additional research is needed. In my opinion, this paragraph should be more related to the results obtained and not limited to very general statements.
